# Estimation of Years Lived with Disability Using a Prevalence-Based Approach: Application to Major Psychiatric Disease in Korea

**DOI:** 10.3390/ijerph18179056

**Published:** 2021-08-27

**Authors:** Chae-Bong Kim, Minsu Ock, Yoon-Sun Jung, Ki-Beom Kim, Young-Eun Kim, Keun-A Kim, Seok-Jun Yoon

**Affiliations:** 1Department of Public Health, Korea University, 73 Goryeodae-ro, Seongbuk-gu, Seoul 02841, Korea; bbp62@naver.com (C.-B.K.); sunnyaurora@nate.com (Y.-S.J.); socommat@korea.ac.kr (K.-B.K.); 2Department of Preventive Medicine, Ulsan University Hospital, University of Ulsan College of Medicine, 877 Bangeojinsunhwando-ro, Dong-gu, Ulsan 44033, Korea; ohohoms@naver.com; 3Department of Big Data Strategy, National Health Insurance Service, 32 Geongang-ro, Wonju 26464, Korea; kimyes4454@gmail.com; 4School of Military Medicine, The Armed Force Medical Command, 90 Jaun-ro, Daejeon 34059, Korea; keunak@korea.ac.kr; 5Department of Preventive Medicine, Korea University College of Medicine, 73 Goryeodae-ro, Seongbuk-gu, Seoul 02841, Korea

**Keywords:** bipolar disorder, burden of disease, major depressive disorder, schizophrenia

## Abstract

To help develop policies concerning the prevention of psychiatric disease in Korea, we reviewed the literature on this topic in different countries and used a prevalence-based approach to estimate the years lived with disability (YLDs) in Korean patients with major psychiatric diseases. We calculated YLDs by extracting data on the number of patients with mild, moderate, and severe cases of schizophrenia, bipolar disorder, and major depressive disorder, as classified by International Statistical Classification of Disease (ICD) codes. YLDs were highest for patients with major depressive disorder (1190.6; 73.9%), schizophrenia (303.3; 18.8%) and bipolar disorder (117.9; 7.3%). Men had higher YLDs for schizophrenia, 2502 (20–24 years); bipolar disorder, 477 (40–44 years); and major depressive disorder, 2034 (75–79 years). Women had higher YLDs for schizophrenia, 484 (45–49 years); bipolar disorder, 214 (≥80 years); and major depressive disorder, 3541 (75–79 years). The prevalence-based approach and severity distribution is useful for estimating long-term psychiatric disease burden and YLDs. However, YLD-estimation studies must compensate for the shortcomings of the ICD-10 by referencing the Diagnostic and Statistical Manual of Mental Disorders 5th edition, as well as updating the disability weight score according to disease severity.

## 1. Introduction

The global burden of disease (GBD) study estimates that 7.4% of disability-adjusted life years (DALYs), 22.9% of years lived with disability (YLD), and 0.5% of years of life lost, globally, are related to mental and substance-use disorders [1]. In terms of disability ranking, major depressive disorder ranked first, schizophrenia was fifth, and bipolar disorder was sixth. Major psychiatric disease accounted for 3.3% of global YLDs [2]. According to the Korea National Burden of Diseases (KNBD) study, DALYs for mental and behavioral disorders accounted for 6.4% among all diseases [3]. Traditionally, the burden of diseases has been identified using an incidence-based approach, based on new diagnostic case units [4]. Since 2012, the Institute for Health Metrics and Evaluation has changed from the incidence-based approach to a prevalence-based approach. In addition, the method by which the disability weight (DW) is calculated was changed from specialized groups to the general population, and the distribution of disease severity has been added. The reduction for future health status and the weight for age were removed [5].

The burden of disease in the population can be calculated using the incidence or prevalence approach [6]. Australia and New Zealand changed their calculation method in the GBD 2010 [7,8] and have used the prevalence-based approach since 2007. While KNBD studies have hitherto calculated YLDs using the incidence-based approach [9,10,11,12], these values have recently been reported according to the prevalence-based approach. Gong et al. reported Korea’s cancer burden using the prevalence-based approach [13]. Park et al. reported Korea’s burden of injury by comparing a prevalence-based approach and an incidence-based approach [14]; the former provided data that were more appropriate for decision-making concerning healthcare resources [13].

In 2016, a KNBD study reported the DALYs of mental and substance-use disorders; however, the results were based on the incidence-based approach [15]. A Brazilian study calculated DALYs of mental and behavioral disorders using the prevalence-based approach and estimated the true global burden of mental illness in 2016 [16,17]. Their data have helped prioritize resource allocation and decision-making related to healthcare policies [18].

Estimating YLDs using the prevalence-based approach can yield data on the prevalence of psychiatric disease, which is needed to develop policies concerning primary or secondary prevention. Therefore, in this study, we set out to review the literature on this topic in different countries and use a prevalence-based approach to estimate the YLDs in Koreans with major psychiatric diseases.

## 2. Materials and Methods

### 2.1. Data Source and Definition of Major Psychiatric Disease

In this study, we used data from the Korean Health Insurance Review and Assessment (HIRA) Services on Healthcare Big-Data Hub. The Healthcare Big-Data Hub includes information on disease prevalence and the use of medical services (http://opendata.hira.or.kr/op/opc/olap3thDsInfo.do accessed on 24 June 2021). The data in the database can be categorized by sex and age, from the age of 5 years. This database has been increasingly used in healthcare research in Korea [19,20].

We obtained the prevalence of patients receiving treatment and diagnosed with specific major psychiatric disease-related codes from January to December 2018 from the Healthcare Big-Data Hub. We defined major psychiatric disease as a main diagnosis of schizophrenia and schizoaffective disorder, bipolar disorder, or major depressive disorder. We reviewed the data using the diagnostic codes (International Statistical Classification of Disease 10th edition [ICD-10]) defined in the GBD 2017 study as causes of death and nonfatal conditions [21]. The ICD-10 codes were as follows: schizophrenia (F20−F20.9), schizoaffective disorder (F25−F25.9), bipolar disorder (F30, F31−F31.9), and depressive disorder and recurrent depressive disorder (F32−F32.9, F33−F33.9).

### 2.2. Literature Review Regarding Major Psychiatric Disease

Diagnostic classifications, such as ICD-10 codes, involve a classification of diseases and clinical syndromes with consideration for the occurrence of disease. As a preliminary study on severity estimation based on the prevalence-based approach, we investigated the rationale for disease classification through the literature. Data were collected from original articles published in academic journals from 1–31 August 2020. The search was performed using the Medical Subject Headings database to select articles meeting the study keywords. The keywords were selected and the search strategy was built according to the participants, intervention, comparison, and outcomes (PICO) criteria. However, we only considered the participants, intervention, and research design in the review. The databases PubMed, Embase, and Google Scholar were searched using the following keyword search algorithm: “mental disorder” or “severe mental illness” or “schizophrenia” or “depressive disorder” or “depressive disorder major” or “bipolar and related disorder” or “bipolar disorder”, and “ICD-10” or “International Classification of Diseases”. The literature was limited to research studies that used ICD-10 codes and subdivisions. The classification of mild, moderate, and severe symptoms in the major psychiatric diseases according to the ICD-10 codes was reviewed twice by two psychiatrists. We confirmed the validity of code mapping in the originally assigned code. Finally, we estimated the severity distribution of major psychiatric diseases based on the classified codes.

### 2.3. Application of Disability Weight

YLDs are calculated by multiplying the number of patients by the DW of those sequelae. The GBD 2010 study used the incidence-based approach, focusing on disease and injury. The development of health state was not necessary since the DW was investigated by experts [22]. However, the prevalence-based approach focuses on the sequelae of disease or injury, allowing for the evaluation of the general population. Therefore, the severity of a condition can be reflected through the health state [23].

When measuring YLDs for mental and behavioral disorders, it is important to apply updated DWs according to disease severity in the country. In Korea, the burden of disease has been estimated using DW, which has recently been updated [23,24]. In this study, we used the health state for severity-specific states of mental illness. Furthermore, we calculated YLDs using the DW reported by Ock et al. [22]: The DW for acute state of schizophrenia was 0.836, while that for the residual state was 0.742; the DW for manic episodes of bipolar disorder was 0.658, while that for the residual state was 0.248; the DW of mild, moderate, and severe episodes of major depressive disorder were 0.551, 0.756, and 0.838, respectively [23]. We applied a DW of 0.742 for mild schizophrenia, 0.789 for moderate schizophrenia (average value of the acute and residual state), and 0.836 for the severe (acute) state. For bipolar disorder, we used a DW of 0.248 for mild, 0.453 for moderate (average value of a manic episode and the residual state), and 0.658 for severe (manic) bipolar disorder. For major depressive disorder, we used a DW of 0.551, 0.756, and 0.838 for mild, moderate, and severe episodes, respectively.

### 2.4. Computation of YLD

To estimate the YLDs of the disease, we used a cause‒sequelae‒health state scheme introduced in the GBD 2010 study. Sequelae are defined as the consequences of diseases, and the health state is designed to reflect the common sequelae [25]. YLDs were calculated by extracting the number of patients, based on ICD-10 code, with diagnoses of schizophrenia and schizoaffective disorder, bipolar disorder, and major depressive disorder. For each disease, the number of patients with mild, moderate, and severe, disease was determined from the ICD-10 codes. Moreover, after reviewing the severity distribution of the disease, we calculated the YLDs per 100,000 individuals according to severity by multiplying the number of patients in each age group and the severity distribution by DW.

## 3. Results

### 3.1. Literature Review of Major Psychiatric Disease According to ICD-10 Codes

The reviewed literature was re-classified according to case and sequelae. Of the nine reports identified, three were that of schizophrenia; two, bipolar disorder; and four, major depressive disorder. The nine selected studies were published between 1999 and 2017. The reported ICD-10 codes in the schizophrenia studies were F20.0, F20.1, F20.2, F20.3, F20.4, F20.5, F20.6, and F25; in the bipolar-disorder studies, they were F30, F31.0, F31.2, F31.3, F31.4, F32.2, and F33.3; and in the studies of major depressive disorder, they were F32.0, F32.1, F32.2, F32.3, F33.0, F33.1, F33.2, F33.3, and F33.4 (Table 1).

### 3.2. Classification Review by Psychiatrists

ICD-10 codes classified for each disease based on a classification review by psychiatrists are shown in Table 2.

### 3.3. Estimation of Severity Distribution of Major Psychiatric Disease

Mild, moderate, and severe symptoms were observed in 13%, 46%, and 41% of the patients receiving treatment for schizophrenia, respectively; in 9%, 61%, and 30% of the patients treated for bipolar disorder, respectively; and in 43%, 40%, and 17% of individuals receiving treatment for major depressive disorder, respectively.

In terms of the proportion of sex and disease severity major psychiatric diseases, schizophrenia (*x*^2^ = 98.123, *p* < 0.0001), bipolar disorders (*x*^2^ = 63.090, *p* < 0.0001), and major depressive disorders (*x*^2^ = 324.542, *p* < 0.0001) showed statistically significant relevance.

Severe schizophrenia involved a higher proportion of females than males, moderate bipolar disorder involved a similar proportion of males and females, and moderate major depressive disorder affected a higher proportion of females than males (Table 3).

### 3.4. Comparison of YLD by Severity of Major Psychiatric Disease

The YLDs for each major psychiatric disease are summarized by disease severity in Table 4. For schizophrenia, the total YLDs (per 100,000 population) were 303.3 YLDs per 100,000 individuals, 279.8 YLDs per 100,000 males, and 326.4 YLDs per 100,000 females. For bipolar disorder, the total YLDs were 117.9 YLDs per 100,000, 97.0 YLDs per 100,000 males, and 138.6 YLDs per 100,000 females. For major depressive disorder, the total YLDs were 1190.6 YLDs per 100,000, 788.4 YLDs per 100,000 males, and 1589.2 YLDs per 100,000 females.

The YLDs due to schizophrenia were higher for females than for males. The YLDs’ gap between males and females was 5.5 YLDs per 100,000 for mild disease, 15.0 YLDs per 100,000 for moderate disease, and 26.1 YLDs per 100,000 for severe disease. Bipolar disorder had higher YLDs in females than in males, and the YLDs’ gap between males and females was 1.1 YLDs per 100,000 for mild, 22.6 YLDs per 100,000 for moderate, and 17.9 YLDs per 100,000 for severe bipolar disease. Major depressive disorder also resulted in higher YLDs in females than in males, and the YLDs’ gap between males and females was 256.6 YLDs per 100,000 for mild, 378.3 YLDs per 100,000 for moderate, and 165.9 YLDs per 100,000 for severe major depressive disorder (Table 4).

### 3.5. Age Distribution of YLD for Each Major Psychiatric Disease among Males

The age distribution of YLDs for each major psychiatric disease among males is shown in Table 5. The YLDs of schizophrenia was the highest (2502 YLDs per 100,000) between the ages of 20–24 years, second highest (2323 YLDs per 100,000) between the ages of 25–29 years, third highest (2240 YLDs per 100,000) between the ages of 45–49 years, and fourth highest (2057 YLDs per 100,000) between the ages of 35–39 years. The YLDs of bipolar disorder was the highest (477 YLDs per 100,000) between the ages of 40–44 years, second highest (462 YLDs per 100,000) between the ages of 45–49 years, third highest (415 YLDs per 100,000) between the ages of 35–39 years, and fourth highest (409 YLDs per 100,000) between the ages of 30–34 years. The YLDs of major depressive disorder was the highest (2034 YLDs per 100,000) between the ages of 75–79 years, second highest (2021 YLDs per 100,000) at ≥80 years, third highest (1651 YLDs per 100,000) between the ages of 70–74 years, and fourth highest (1266 YLDs per 100,000) between the ages of 65–69 years (Table 5).

### 3.6. Age Distribution of YLD for Each Major Psychiatric Disease among Females

The age distribution of YLDs for each major psychiatric disease among females is shown in Table 6. YLDs of schizophrenia was the highest (484 YLDs per 100,000) between the ages of 45–49 years, followed by 461 YLDs per 100,000 between the ages of 50–54 years, 455 YLDs per 100,000 between the ages of 40–44 years, and 455 YLDs per 100,000 between the ages of 30–34 years. The YLDs of bipolar disorder was the highest at 214 YLDs per 100,000 in the ≥80-years group, second highest (202 YLDs per 100,000) between the ages of 25–29 years, third highest (196 YLDs per 100,000) between the ages of 30–34 years, and fourth highest (182 YLDs per 100,000) between the ages of 20–24 years. The YLDs of major depressive disorder was the highest (3541 YLDs per 100,000) between the ages of 75–79 years, second highest (3259 YLDs per 100,000) between the ages of 70–74 years, third highest (2795 YLDs per 100,000) between the ages of 65–69 years, and fourth highest (2387 YLDs per 100,000) at ≥80 years (Table 6).

## 4. Discussion

Schizophrenia, depressive disorder, and anxiety disorder account for more than 70% of DALYs [35], indicating these psychiatric conditions as among the most damaging [36]. Using a prevalence-based approach, we confirmed that the YLDs of major psychiatric disease was high. The severity distribution analysis of major psychiatric disease revealed that most schizophrenic patients had moderate (46%) or severe (41%) disease, while only a minority had mild (13%) schizophrenia; almost two-thirds of the cases of bipolar disorder were moderate (61%), approximately one-third was severe (30%), and a small proportion was mild (9%); major depressive disorder was mostly mild (43%) or moderate (40%), while some patients had severe (17%) cases of the condition. In the severity and sex distribution analysis, schizophrenia and bipolar disorder were higher among females than in males, while moderate depressive disorders were higher among females. In the sex and severity proportion of major psychiatry diseases, schizophrenia, bipolar disorder, and major depressive disorders showed statistically significant relevance. In the prevalence study, schizophrenia was higher among females than males, and the prevalence of schizophrenia was higher in the age range of 45–54 years and 35–44 years [37]. The prevalence of bipolar disorder was higher among females (0.23%) than males (0.17%) [38]. Further, the symptoms of depression were more severe among females than males [39]. Additionally, a previous study that reported on the Diagnostic and Statistical Manual-5th edition (DSM-5) for evaluation of the severity of schizophrenia and bipolar disorder showed a severity proportion similar to that in our study [40]. The most widely used diagnostic manuals are the ICD-10 and the DSM-5, and these form the basis for mental disease diagnoses in most parts of the world. However, the comparison between the two manuals is limited because severity assessments should take into account the course of the disease and the level of treatment for the individual. YLDs for major depressive disorder were highest, accounting for 73.9% (1190.6 YLDs per 100,000) of total YLDs for major psychiatric disease, followed by schizophrenia at 18.8% (303.3 YLDs per 100,000), and bipolar disorder at 7.3% (117.9 YLDs per 100,000).

Among individuals with schizophrenia, moderate and severe cases accounted for 87% of the total population. Both males and females had higher YLDs in their 40s and 50s. Patients with schizophrenia reported a high rate of treatment discontinuation, low recovery rates, and difficulty in independent living [41]. In terms of the economic burden of schizophrenia in Korea, both males and females had the highest YLDs in the 40–49-year age group. Furthermore, the total death attributable to patients with schizophrenia in this age group was the highest [42]. Although this study reported higher YLDs in women than in men, no conclusions can be drawn from this finding, as sex-based differences in schizophrenia may vary depending on disease progression, hormonal differences, age, and sex behavior patterns [43]. In a Korean study on bipolar disorder, the prevalence was higher among females, and the rate of increase in its prevalence was high for those aged ≥60 years [38]. In another study on bipolar disorder in 2013, females had higher prevalence rates than males, accounting for 0.4% of the total DALY, 1.3% of total YLD, and DALYs in the 20–50 year age group [44]. In this investigation, we observed that a high proportion of individuals with bipolar disorder were moderately affected and that YLDs were higher among older adults (≥80 years). Bipolar disorder involves repeated episodes of mania and depression and is likely to recur [45]. In addition, continuous treatment and care can lead to the worsening of symptoms and are associated with suicide attempts [46]. Major depressive disorder had the highest YLDs among major psychiatric diseases, although the greatest proportion of patients had mild or moderate cases of the condition. Several previous studies reported a relationship between low quality of life and major depressive disorder. We confirmed that major depressive disorder mostly involved cases of low severity, but high YLDs. We surmise that this reflects a disease-induced low quality of life over a long period of time. Indeed, this finding reflects the utility of the prevalence-based approach in measuring the long-term burden of diseases [13].

Vigo et al. reported that approaches to diagnostic classification, such as the ICD-10, as methods for estimating mental illness need to consider the clinical syndrome and incidence [17]. In psychiatry, the patient’s behavior and abnormal experience are diagnosed by subjective reports by the physician [47]. In this study, the severity classification using ICD-10 codes was evaluated by psychiatrists. However, classification codes determined through evaluation by physicians may have had a limited impact on the results due to underestimation. In addition, since most psychiatrists in Korea have been educated and trained with the Diagnostic and Statistical Manual-5th edition (DSM-5), the accuracy of assigning ICD-10 codes is unlikely to be high. This means that the severity of the patient’s condition may differ from that indicated by the ICD-10 code. We speculate that the psychiatrists categorized the severity using ICD-10 codes to match the patient symptoms as closely as possible; however, in general, the severity of diseases changes with diagnostic codes according to the time point of the episode of diagnostic criteria. Therefore, this study was limited by the difficulty of accurately estimating the severity distribution among the psychiatric patients.

The DSM-5 diagnoses of patients with schizophrenia, bipolar disorder, and major depressive disorder have been validated in the reported Korean populations: 79.0%, 89.3%, and 86.9% for major depressive disorder, bipolar disorder, and schizophrenia, respectively [48]. This implies that symptom changes in patients with major psychiatric disease are not significant. In comparing the ICD-10 and DSM-5 for bipolar disorder, both systems showed some consistency in the diagnosis of bipolar disorder and related diseases. However, direct comparisons between the ICD-10 and DSM-5 were limited because there was a time gap of 20 years and a distinct difference in criteria, classification, and details between the systems [47]. Although, we point to a realistic problem arising from the difference between DSM-5 and ICD-10, the use of ICD-10 is not a major factor affecting the diagnoses of schizophrenia, bipolar disorder, and major depressive disorder. Therefore, we propose a future follow-up study that combines DSM-5 and ICD-10 to clearly classify patients’ symptoms. Further, this study stresses the importance of physicians adequately recording the patients’ symptoms to increase the reliability of YLD results using a prevalence-based approach.

DW is an important value for calculations in the prevalence-based approach. We used the DW reported by Ock et al. to calculate the YLDs for major psychiatric disease [22]. However, we applied the mean values for mild and severe DWs to moderate severity because the DW for moderate disease had not been reported. Recently, a study from Thailand reported DW to estimate the burden of disease in patients with major depressive disorder and alcohol disorder [49]. In future research, more detailed YLD calculations could be conducted to calculate the severity of DW separately for mental illnesses.

The severity of major psychiatric disease did not change significantly and is assumed to be maintained from the onset of the disease until death. In this study, we considered literature as a basis for the classification of major psychiatric disease severity, and psychiatrists reclassified ICD-10 codes based on the literature. However, major psychiatric disease can be assessed more reasonably through prevalence duration, drug use, and the number of hospitalizations. Future studies will require an evaluation that reflects the indicators affecting severity.

No study to date has classified the severity of major psychiatric disease according to the ICD-10 codes. The prevalence-based approach used in this study confirmed the severity distribution within the estimates under this disease classification system. We expect that this study will serve as a basis for future observation and evaluation of the epidemiological distribution of major psychiatric disease.

## 5. Conclusions

The GBD is a very important issue, and its significance will be emphasized more in the future. The prevalence-based approach was useful for estimating major psychiatric diseases’ burden in South Korea. YLD estimation studies need to verify the credibility of clinical experts’ ICD-10 coding and carefully classify patients’ symptoms by matching ICD-10 and DSM-5 criteria

## Figures and Tables

**Table 1 ijerph-18-09056-t001:** Literature review of major psychiatric diseases using ICD-10 codes.

Major Psychiatric Disease	Country	Study Design (Year)	Participants (N)	Age Ranges and Means (Years)	ICD-10 Codes	Reference
Schizophrenia	Germany	Retrospective cohort study (1993–1997)	Cases 126	35.8	F20.0, F20.1 F20.2, F20.3, F20.5, F20.6, F25	[26]
Schizophrenia	Mexico	Case-control study (2009–2010)	Cases 50 Controls 150	18–72 45.1	F20.0, F20.2, F20.3, F20.4, F20.5, F20.6	[27]
Schizophrenia	Sweden	Case-control study	Cases 19 Controls 92	18–65	F20.0–F20.9 F25.0–F25.9	[28]
Bipolar disorder	Germany	Double-blind study	Cases 20 Placebo 20	18–65	F31.2, F31.3, F31.4	[29]
Bipolar disorder	Sweden	Retrospective cohort study (1973–2007)	Cases 45,087 (male)	18–20	F30, F31.0	[30]
Major depressive disorder	Germany	Double-blind study, follow-up	Cases 216 Placebo 47	18–65	F32.0, F32.1, F32.2, F32.3, F33.0, F33.1, F33.2, F33.3	[31]
Major depressive disorder	-	Case-control study	Cases 60	41	F33.1, F33.2	[32]
Major depressive disorder	England	Randomized controlled trials (1998–2005)	Depression cases 105	-	F32.3	[33]
Major depressive disorder	Switzerland	Prospective clinical study	Cases 47 Control 110	24.86–26.86	F33.4, F32.0, F33.0, F32.1, F33.1, F32.2, F32.3, F33.2, F33.3	[34]

ICD-10 = International Classification of Diseases, 10th edition.

**Table 2 ijerph-18-09056-t002:** Mapping of ICD-10 code-based severity classification.

Major Psychiatric Disease	Mild	Moderate	Severe
Schizophrenia	F20.2, F20.4, F20.5, F20.6, F20.8, F20.9, F25.8, F25.9	F20.0, F20.3, F20.8, F20.9, F25.8, F25.9	F20.0, F20.1, F25.0, F25.1, F25.2, F20.8, F20.9, F25.8, F25.9
Bipolar disorders	F31.7, F30.8, F30.9, F31.8, F31.9	F30.0, F31.0, F31.1, F31.3, F31.6, F30.8, F30.9, F31.8, F31.9	F30.1, F30.2, F31.2, F31.4, F31.5, F30.8, F30.9, F31.8, F31.9
Major depressive disorders	F32.0, F33.0, F33.4, F32.8, F32.9, F33.8, F33.9	F32.1, F33.1, F32.8, F32.9, F33.8, F33.9	F32.2, F32.3, F33.2, F33.3, F32.8, F32.9, F33.8, F33.9

ICD-10 = International Classification of Diseases, 10th edition.

**Table 3 ijerph-18-09056-t003:** Proportion of disease severity and sex in each group of major psychiatric diseases using ICD-10 codes in Korea.

Major Psychiatric Disease	Schizophrenia			Bipolar Disorder			Major Depressive Disorder		
	Mild N (%)	Moderate N (%)	Severe N (%)	Mild N (%)	Moderate N (%)	Severe N (%)	Mild N (%)	Moderate N (%)	Severe N (%)
Males	11,565 (13)	41,321 (47)	35,029 (40)	4550 (9)	29,977 (61)	14,738 (30)	128,773 (44)	114,179 (39)	49,181 (17)
Females	13,253 (13)	46,641 (45)	43,491 (42)	5616 (8)	43,085 (61)	21,725 (31)	249,173 (42)	241,583 (41)	99,898 (17)
Males + Females	24,818 (13)	88,034 (46)	78,445 (41)	10,528 (9)	73,062 (61)	36,103 (30)	379,054 (43)	354,672 (40)	149,080 (17)
*x*^2^ (*p*) ^1^	98.123 (0.0001)			63.090 (0.0001)			324.542 (0.0001)		

ICD-10 = International Classification of Disease, 10th edition. ^1^ Values are expressed as the chi square.

**Table 4 ijerph-18-09056-t004:** YLD rates in Korea attributable to the severity of major psychiatric disease according to sex.

	YLD Rates of Schizophrenia ^1^			YLD Rates of Bipolar Disorder ^1^			YLD Rates of Major Depressive Disorder ^1^		
	Mild	Moderate	Severe	Mild	Moderate	Severe	Mild	Moderate	Severe
Male + Female	36.5	137.0	129.8	4.9	65.4	47.6	410.1	532.0	248.5
Male	33.7	129.4	116.7	4.4	54.0	38.6	281.2	342.0	165.2
Female	39.2	144.4	142.8	5.5	76.6	56.5	537.8	720.3	331.1

YLDs = years lived with disability, ^1^ per 100,000 population.

**Table 5 ijerph-18-09056-t005:** Prevalence-based approach YLDs and YLD rates for major psychiatric disease among males by age.

	Schizophrenia		Bipolar Disorder		Major Depressive Disorder	
Age Groups, Years	YLDs	YLD Rates ^1^	YLDs	YLD Rates ^1^	YLDs	YLD Rates ^1^
0–4	5	1	0	0	36	4
5–9	24	53	5	2	500	43
10–14	121	209	18	10	2061	176
15–19	1512	1108	77	106	10,300	720
20–24	4475	2502	145	259	17,895	1034
25–29	6314	2323	135	366	13,195	766
30–34	6533	1905	119	409	11,340	710
35–39	8378	2057	102	415	13,411	664
40–44	9217	2009	104	477	13,230	685
45–49	10,394	2240	100	462	15,058	669
50–54	7671	1928	92	366	14,240	680
55–59	5682	1945	92	269	16,558	783
60–64	3917	1660	97	229	16,236	950
65–69	2531	1319	116	223	14,392	1266
70–74	1588	1075	125	185	14,194	1651
75–79	1154	1023	149	168	13,941	2034
≥80	964	1080	182	162	11,999	2021

YLDs = years lived with disability, ^1^ per 100,000 population.

**Table 6 ijerph-18-09056-t006:** Prevalence-based approach YLDs and YLD rates for major psychiatric disease among females by age.

	Schizophrenia		Bipolar Disorder		Major Depressive Disorder	
Age Groups, Year	YLDs	YLDs Rates ^1^	YLDs	YLDs Rates ^1^	YLDs	YLDs Rates ^1^
0–4	1	0	2	0	23	2
5–9	10	1	10	1	283	26
10–14	176	16	242	22	4572	417
15–19	1367	104	1353	103	16,883	1284
20–24	3596	226	2897	182	24,029	1511
25–29	6008	379	3198	202	22,859	1444
30–34	6838	455	2947	196	21,514	1432
35–39	8309	428	3335	172	25,923	1335
40–44	8521	455	3055	163	25,029	1336
45–49	10,636	484	3210	146	28,052	1276
50–54	9527	461	2816	136	29,746	1439
55–59	9260	439	2765	131	38,482	1823
60–64	7088	406	2353	135	38,132	2182
65–69	4290	352	1710	140	34,075	2795
70–74	2836	286	1376	139	32,264	3259
75‒79	1934	208	1353	145	32,978	3541
≥80	2583	212	2601	214	29,069	2387

YLDs = years lived with disability, ^1^ per 100,000 population.

## Data Availability

Publicly available data sets were analyzed in this study. This data can be found here: [http://opendata.hira.or.kr/op/opc/olap3thDsInfo.do accessed on 24 June 2021].

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
