# Peer review of "Estimation of Years Lived with Disability Using a Prevalence-Based Approach: Application to Major Psychiatric Disease in Korea"

_ijerph, 2021, doi:10.3390/ijerph18179056_

Round 1
Reviewer 1 Report
The paper show a method based on prevalence to estimate the Years Lived with Disability (YLD) for psychiatric diseases. This approach could be of interest in order to engage preventive interventions. This work has two additional aspects, one involving the analysis by disease severity and, the gender comparisons.
The paper is well-constructed and it is understandable. However, I have two main points.
First, data obtaines is not discussed according the literature existing (i.e. gender differences, severity of the disease,...) and, second there are no statistical analysis that support some affirmations (gender or severity differences).
Author Response
Thank you for giving us the opportunity to revise and resubmit our manuscript titled, “Estimation of Years Lived with Disability Using a Prevalence-Based Approach: Application to Major Psychiatric Disease in Korea” to the International Journal of Environmental Research and Public Health. The manuscript id is ijerph-1273945.
We are grateful to you and the reviewers for your careful and thoughtful comments, and we have made every attempt to fully address those comments in the revised manuscript. Our manuscript has certainly benefited from your insightful suggestions. We look forward to working with you and the reviewers to move this manuscript closer to publication in the Journal of International Journal of Environmental Research and Public Health. Below, we have provided the point-by-point responses to the reviewers’ comments.
We hope that the revised manuscript is now acceptable for publication. If you have any further suggestions or requests, we will be glad to address them.
Sincerely,
Seok-Jun Yoon, MD, PhD
Dear reviewer 1
We appreciate your interest in the YLD estimation study on major psychiatric diseases in Korea. We have been able to improve our research through your insightful comments, for which, we are grateful. The purpose of this study was to use a prevalence-based approach to estimate the YLD for schizophrenia, bipolar disorders, and major depressive disorders in South Korea.
- Response to reviewer1 comments:
- (1) Review report: First, data obtaines is not discussed according the literature existing (i.e. gender differences, severity of the disease,...)
- (2) Second there are no statistical analysis that support some affirmations (gender or severity differences).
- Response to review report (1):
- In this study, we estimated the severity of major psychiatric diseases and also performed sex and age comparisons. However, we had initially not included the details regarding differences in sex, age difference, and disease severity in the Discussion. Therefore, we have reviewed the existing literature and supplemented the Discussion section with more information on differences in disease severity, sex, and age in major psychiatric diseases.
- Revised in manuscript: page 8, line 230-245.
- In the severity and sex distribution analysis, schizophrenia and bipolar disorder were higher among females than in males, while moderate depressive disorders were higher among females. In the sex and severity proportion of major psychiatry diseases, schizophrenia, bipolar disorder, and major depressive disorders showed statistically significant relevance. In the prevalence study, schizophrenia was higher among females than males, and the prevalence of schizophrenia was higher in the age range of 45−54 years and 35−44 years [37]. The prevalence of bipolar disorder was higher among females (0.23%) than males (0.17%) [38]. Further, the symptoms of depression were more severe among females than males [39]. Additionally, a previous study that reported on the Diagnostic and Sta-tis-tical Manual-5th edition (DSM-5) for evaluation of the severity of schizophrenia and bipolar disorder has shown a severity proportion similar to that in our study [40]. The most widely used diagnostic manuals are the ICD-10 and the DSM-5, and these form the basis for mental disease diagnoses in most parts of the world. However, the comparison be-tween the two manuals is limited because severity assessments should take into ac-count the course of the disease and the level of treatment for the individual.
- Response to review report (2):
- Although we included the proportion of sex and severity differences in major psychiatric diseases in Table 3, we had not included the statistical analysis results. Therefore, Table 3 has been revised to include the statistical analysis results in addition to the prevalence of sex and severity differences in psychiatric diseases. The statistical analysis method used in this study was crossover analysis, and statistical significance was evaluated using Chi-square test.
- Revised in manuscript: page 5, line 157-165.
- In terms of the proportion of sex and disease severity major psychiatric diseases, schizophrenia (x2=98.123, p<0.0001), bipolar disorders (x2=63.090, p<0.0001), and major depressive disorders (x2=324.542, p<0.0001) showed statistically significant relevance.
- Table 3. Proportion of disease severity and sex in each group of major psychiatric disease using ICD−10 codes in Korea
|
Major psychiatric disease |
Schizophrenia |
Bipolar disorder |
Major depressive disorder |
||||||
|
Mild N (%) |
Moderate N (%) |
Severe N (%) |
Mild N (%) |
Moderate N (%) |
Severe N (%) |
Mild N (%) |
Moderate N (%) |
Severe N (%) |
|
|
Males |
11,565 (13) |
41,321 (47) |
35,029 (40) |
4,550 (9) |
29,977 (61) |
14,738 (30) |
128,773 (44) |
114,179 (39) |
49,181 (17) |
|
Females |
13,253 (13) |
46,641 (45) |
43,491 (42) |
5,616 (8) |
43,085 (61) |
21,725 (31) |
249,173 (42) |
241,583 (41) |
99,898 (17) |
|
Males + Females |
24,818 (13) |
88,034 (46) |
78,445 (41) |
10,528 (9) |
73,062 (61) |
36,103 (30) |
379,054 (43) |
354,672 (40) |
149,080 (17) |
|
x2 (p) 1 |
98.123 (.0001) |
63.090 (.0001) |
324.542 (.0001) |
||||||
ICD−10 = International Classification of Disease, 10th edition
- Values are expressed as the chi square

Reviewer 2 Report
This study refers to a very important topic of disability assiciated with major psychiatric disorders. However the manuscrpt needs some improvements:
I. Line 85:
The Authors list major depressive disorder with the ICD-10 codes: (F32−F32.9, F33−F33.9). I would like to ask the Authors to be more precise. More exactly these codes refer to Depressive didorder and Recurrent depressive disorder
II. As far as I understood the idea of the study, it was divided into 2 parts:
- Revision of literature in the topic in different countries (Results 3.1).
- Analysis of the problem among patients bin Korea (Results 3.3).
In my understanding is correct, that shold be mentioned in the objective of the study and in the abstract. And in the part concerning the literature review, please specify, what method of review was adopted, PRISMA guidelines, PICO method. Etc. ?
III.There is a double dot in the sentence:
„When measuring YLDs for mental and behavioral disorders, it is important to apply 106
updated DWs according to disease severity in the country. . In Korea, the burden of dis- 107
ease has been estimated using DW, which have recently been updated [23,24].”
Author Response
Thank you for giving us the opportunity to revise and resubmit our manuscript titled, “Estimation of Years Lived with Disability Using a Prevalence-Based Approach: Application to Major Psychiatric Disease in Korea” to the International Journal of Environmental Research and Public Health. The manuscript id is ijerph-1273945.
We are grateful to you and the reviewers for your careful and thoughtful comments, and we have made every attempt to fully address those comments in the revised manuscript. Our manuscript has certainly benefited from your insightful suggestions. We look forward to working with you and the reviewers to move this manuscript closer to publication in the Journal of International Journal of Environmental Research and Public Health. Below, we have provided the point-by-point responses to the reviewers’ comments.
We hope that the revised manuscript is now acceptable for publication. If you have any further suggestions or requests, we will be glad to address them.
Sincerely,
Seok-Jun Yoon, MD, PhD
Dear reviewer 2
We appreciate your interest in the YLD estimation study on major psychiatric diseases in Korea. We have been able to improve our research through your insightful comments, for which, we are grateful. The purpose of this study was to estimate the YLD for schizophrenia, bipolar disorders, and major depressive disorders in South Korea using a prevalence-based approach.
- Response to reviwer2 comments:
- (1) Review report: The Authors list major depressive disorder with the ICD-10 codes: (F32−F32.9, F33−F33.9). I would like to ask the Authors to be more precise. More exactly these codes refer to Depressive disorder and Recurrent depressive disorder.
- (2) In my understanding is correct, that should be mentioned in the objective of the study and in the abstract.
- (3) In the part concerning the literature review, please specify, what method of review was adopted, PRISMA guidelines, PICO method. Etc. ?
- (4) There is a double dot in the sentence on line 106-107.
- Response to review report (1):
- We agree with you; therefore, we have accurately represented the code names for major depressive disorders. We have classified these into depressive disorder and recurrence depressive disorder on line 85.
- Revised in manuscript: page 2, line 83-85.
- The ICD-10 codes were as follows: schizophrenia (F20−F20.9), schizoaffective disorder (F25−F25.9), bipolar disorder (F30, F31−F31.9), and depressive disorder and recurrent de-pressive disordermajor depressive disorder (F32−F32.9, F33−F33.9).
- Response to review report (2):
- The purpose of this study was to estimate the YLD for patients with major psychiatric diseases in South Korea and compare sex and age differences among them using a prevalence-based approach. We reviewed each country’s subjective literature to achieve our research purpose, evaluated the disease severity classification, and included the disease severity in patients with major psychiatric diseases in Table 3. Therefore, we have included the research ideas you understood in the purpose and abstract of the study.
- Revised in manuscript: page 1, line 18-20.
- To help develop policies concerning the prevention of psychiatric disease in Korea, we reviewed the literature on this topic in different countries and used a prevalence-based approach to estimate the years lived with disability (YLDs) in Korean patients with major psychiatric diseases.
- Revised in manuscript: page 2, line 66-68.
- Therefore, in this study, we set out to review the literature on this topic in different coun-tries and use a prevalence-based approach to estimate the YLDs in Koreans with major psychiatric diseases.
- Response to review report (3):
- We recognized that a literature basis was needed to assess the severity of major psychiatric diseases in Korea. We have reviewed previous literature to classify the severity of major mental illnesses, and crudely categorized ICD codes by organizing the literature reviewed. Psychiatric experts reviewed the literature information to classify and evaluate the ICD-10 codes. We used the Patient, Intervention, Comparison, Outcome (PICO) method to review the ICD-10 codes mentioned in the articles of interest. However, we did not include (C) comparisons and (O) outcomes in the review. We only considered (P) participants, (I) intervention, and research design. We reviewed the literature by referring to the preference reporting topics in the Systematic Review and Meta Analysis 2009 Checklist to ensure that the literature on the topics of interest was included.
- Revised in manuscript: page 2-3, line 91-104.
- The search was performed using the Medical Subject Headings database to select article meeting the study keywords. The keywords were selected, and the search strategy was built according to the participants, intervention, comparison, and outcomes (PICO) criteria. However, we only considered the participants, intervention, and research design in the review. The databases PubMed, Embase, and Google Scholar were searched using follow-ing keyword search algorithm: “mental disorder” or “severe mental illness” or “schizophrenia” or “depressive disorder” or “depressive disorder major” or “bipolar and related disorder” or “bipolar disorder”, and “ICD-10” or “International Classification of Dis-eases”. The literature was limited to research studies that used ICD-10 codes and subdivisions. The classification of mild, moderate, and severe symptoms in the major psychiat-ric disease according to the ICD-10 codes was reviewed twice by two psychiatrists. We confirmed the validity of code mapping in the originally assigned code. Finally, we estimated the severity distribution of major psychiatric diseases based on the classified codes.
- Response to review report (4):
- We found a typing error on line 106-107 and corrected it.
- Revised in manuscript: page 3, line 126-127.
- We have corrected the typing error.
- These references have been included in the manuscript.
- Revised in manuscript: page 12, line 435-445.
- Cho, S.J.; Kim, J.; Kang, Y.J.; Lee, S.Y.; Seo, H.Y.; Park, J.E.; Kim, H.; Kim, K.-N.; Lee, J.Y.; Sohn, J.H. Annual prevalence and incidence of schizophrenia and similar psychotic disorders in the Republic of Korea: a National Health Insurance data-based study. Psychiatry Invest 2020, 17(1), 61-70
- Jung, Y.-S.; Kim, Y.-E.; Kim, A.; Yoon, S.-J. Trends in the prevalence and treatment of bipolar affective disorder in South Korea. Asian J Psychiatr 2020, 53, 102194.
- Kim, J.-H.; Cho, M.J.; Hong, J.P.; Bae, J.N.; Cho, S.-J.; Hahm, B.-J.; Lee, D.-W.; Park, J.-I.; Lee, J.-Y.; Jeon, H.J.; Chang, S.M. Gender differences in depressive symptom profile: results from nationwide general population surveys in Korea. J Korean Med Sci 2015, 30(11), 1659-1666.
- Sagar, R.; Pattanayak, R.D.; Chandrasekaran, R.; Chaudhury, P.K.; Deswal, B.S.; Singh, R.K.L.; Malhotra, S.; Nizamie, S.H.; Panchal, B.N.; Sudhakar, T.P.; Trivedi, J.K.; Varghese, M.; Prasad, J.; Chatterji, S. Twelve-month prevalence and treatment gap for common mental disorders: findings from a large-scale epidemiological survey in India. Indian J Psychiatry 2017, 59(1), 46-55.

Round 2
Reviewer 2 Report
I would like to thank the Authors for all changes and improvements they have done. I have no additional comments.
Author Response
August 24, 2021
Prof. Dr. Paul B. Tchounwou
Editor-in-Chief
International Journal of Environmental Research and Public Health
Dear Editor:
Thank you for the opportunity to revise and resubmit our manuscript titled “Estimation of Years Lived with Disability Using a Prevalence-Based Approach: Application to Major Psychiatric Disease in Korea” to the International Journal of Environmental Research and Public Health. The manuscript id is ijerph-1273945.
We are grateful to you and the academic editor and reviewer for your thoughtful comments. Our manuscript has certainly benefited from reviewer and academic editor insightful comments. We have provided the responses to the comments of academic editor. We hope that the revised manuscript is now acceptable for publication. If you have any further suggestions or requests, we will be glad to address them.
Sincerely,
Seok-Jun Yoon, MD, PhD
Dear reviewer
Thank you for considering my article for publication in the International Journal of Environmental Research and Public Health. We are grateful to you for the valuable comments provided. I would be happy to make any further changes that may be required. Here are responses to the comments:
- The Reviewer’s Comments
- I would like to thank the Authors for all changes and improvements they have done. I have no additional comments.
- Responses to the Reviewer’s Comments:
- Thank you for your insightful comments and for the opportunity to resubmit our manuscript "Estimation of Years Lived with Disability Using a Prevalence-Based Approach: Application to Major Psychiatric Disease in Korea" to the International Journal of Environmental Research and Public Health.